# *Porphyromonas gingivalis* Outer Membrane Vesicles Stimulate Gingival Epithelial Cells to Induce Pro-Inflammatory Cytokines via the MAPK and STING Pathways

**DOI:** 10.3390/biomedicines10102643

**Published:** 2022-10-20

**Authors:** Yuta Uemura, Yuka Hiroshima, Ayano Tada, Keiji Murakami, Kaya Yoshida, Yuji Inagaki, Tomomi Kuwahara, Akikazu Murakami, Hideki Fujii, Hiromichi Yumoto

**Affiliations:** 1Department of Periodontology and Endodontology, Graduate School of Biomedical Sciences, Tokushima University, 3-18-15 Kuramoto, Tokushima 770-8504, Japan; 2Department of Oral Microbiology, Graduate School of Biomedical Sciences, Tokushima University, 3-18-15 Kuramoto, Tokushima 770-8504, Japan; 3Department of Microbiology, Faculty of Medicine, Kagawa University, 1750-1 Miki, Takamatsu 761-0793, Japan; 4Department of Clinical Nutrition, Faculty of Health Science and Technology, Kawasaki University of Medical Welfare, 288 Matsushima, Kurashiki 701-0193, Japan; 5Department of Oral Healthcare Education, Graduate School of Biomedical Sciences, Tokushima University, 3-18-15 Kuramoto, Tokushima 770-8504, Japan; 6Department of Biology, Keio University School of Medicine, 4-1-1 Hiyoshi, Kohoku-ku, Yokohama 223-8521, Japan

**Keywords:** *Porphyromonas gingivalis*, outer membrane vesicles, stimulator of interferon genes, inflammation, periodontitis

## Abstract

*Porphyromonas gingivalis* (*Pg*) is a keystone pathogen associated with chronic periodontitis and produces outer membrane vesicles (OMVs) that contain lipopolysaccharide (LPS), gingipains, and pathogen-derived DNA and RNA. *Pg*-OMVs are involved in the pathogenesis of periodontitis. *Pg*-OMV-activated pathways that induce the production of the pro-inflammatory cytokines, interleukin (IL)-6, and IL-8 in the human gingival epithelial cell line, OBA-9, were investigated. The role of mitogen-activated protein kinase (MAPK) and nuclear factor (NF)-κB in levels of *Pg*-OMV-induced pro-inflammatory cytokines was investigated using Western blot analysis and specific pathway inhibitors. *Pg*-OMVs induced IL-6 and IL-8 production via the extracellular signal-regulated kinase (Erk) 1/2, c-Jun N-terminal kinase (JNK), p38 MAPK, and NF-κB signaling pathways in OBA-9 cells. In addition, the stimulator of interferon genes (STING), an essential innate immune signaling molecule, was triggered by a cytosolic pathogen DNA. *Pg*-OMV-induced IL-6 and IL-8 mRNA expression and production were significantly suppressed by STING-specific small interfering RNA. Taken together, these results demonstrated that *Pg*-OMV-activated Erk1/2, JNK, p38 MAPK, STING, and NF-κB signaling pathways resulting in increased IL-6 and IL-8 expression in human gingival epithelial cells. These results suggest that *Pg*-OMVs may play important roles in periodontitis exacerbation by stimulating various pathways.

## 1. Introduction

Periodontitis is a chronic inflammatory disease caused by periodontopathogenic bacteria in dental plaques [1]. Continuous stimulation and an increase in the number of periodontopathogenic bacteria cause inflammatory responses, including pro-inflammatory cytokine production that is executed by host innate immune responses in periodontal tissue, leading to bone and tissue destruction, and eventually tooth loss [1]. In addition, chronic periodontitis is associated with various systemic diseases, such as cardiovascular disease, diabetes, rheumatoid arthritis, and kidney disease [2]. *Porohyromonas gingivalis* (*Pg*) is a keystone pathogen associated with chronic periodontitis and contributes to systemic diseases [3]. *Pg* expresses several virulence factors, such as lipopolysaccharide (LPS), capsular polysaccharide, fimbriae, and gingipains (cysteine proteinases) [4]. *Pg*-LPS strongly aggravates periodontitis through stimulation of intracellular signaling events by the host immune system, such as the induction of pro-inflammatory cytokines, chemokines, and other inflammation-related molecules in periodontal tissue [4]. 

Both pathogenic and commensal Gram-negative bacteria can produce outer membrane vesicles (OMVs) and release them into the extracellular environment [5]. OMVs contain LPS, periplasmic and membrane-bound proteins, enzymes, toxins, DNA, RNA, and peptidoglycan [5]. OMVs have been proposed to be involved in cell–cell interactions, nutrient acquisition, host immune dysregulation and modulation, host cell interaction, and biofilm formation [5]. Several studies using proteomic tools have validated that *Pg*-OMVs contain various virulence factors, including FimA, HagA, mating factors, and gingipains, which are involved in periodontitis pathology [6,7,8]. Since *Pg*-OMVs are smaller than bacterial cells, *Pg*-OMVs can be transported rapidly through the blood circulation and therefore affect distant tissues and organs [9], resulting in the modulation of various responses. For instance, *Pg*-OMVs containing gingipains accumulate in the liver and inhibit insulin-stimulated glycogen synthesis [8]. Moreover, *Pg*-OMVs induce apoptosis and destroy the lung epithelial cell barrier [10]. These results suggest that gingipains in *Pg*-OMVs play an important role in systemic diseases related to *Pg* infection.

*Pg*-OMVs penetrate epithelial cells in a fimbria-dependent manner and are endocytosed through actin-dependent lipid raft-mediated pathways [11]; further, they cause host cells to produce a variety of inflammatory factors [12]. Moreover, *Pg*-OMVs induce nuclear factor (NF)-κB (NF-κB) activation, promote apoptosis and glycolysis, and inhibit proliferation in epithelial cells [12]. Therefore, *Pg*-OMVs contribute to host cell invasion, host cell destruction, escape from the immune system, and antibiotic resistance therefore leading to inflammation induction in gingival tissue, and thereby facilitating the progression of periodontitis [12].

Stimulator of interferon genes (STING) is a critical and essential innate immune signaling adaptor molecule for controlling the transcription of numerous host defense genes, including type I interferon (IFNs) and inflammatory cytokines. STING is activated by recognizing cyclic GMP–AMP (cGAMP), which is synthesized by the cytoplasmic enzyme—cyclic GMP–AMP synthase (cGAS)—in response to pathogen-derived DNA [13,14,15]. In periodontitis, a strong STING accumulation was observed in the basal epithelium and around vessel walls in the connective tissue, however, STING was also weakly present in healthy gingiva [16]. Hence, we considered that the STING-mediated pathway may play an important role in the induction of inflammation by *Pg*-OMV-derived DNA in periodontitis.

Based on these findings, we hypothesized that gingipains and DNA in *Pg*-OMVs may contribute to the pathogenesis and the progression of periodontitis. Since the molecular mechanisms underlying the effects of *Pg*-OMVs on gingival epithelial cells and the establishment of local infection remain obscure, the possible mechanisms underlying *Pg*-OMV-mediated gingival epithelial cell events including the STING-mediated pathway were investigated.

## 2. Materials and Methods

### 2.1. Bacterial Cultures

*Pg* ATCC33277 (type strain) was cultured anaerobically in a brain heart infusion (BHI) broth (Becton Dickinson, Sparks, MD, USA) supplemented with 5 μg/mL hemin and 1 μg/mL menadione for 2 days. Gingipain mutant (Δ*rgpA*, Δ*rgpB*, and Δ*kgp*) of *Pg* ATCC33277 (KDP136 strain) was kindly provided by Prof. Koji Nakayama (Nagasaki University Graduate School of Biomedical Sciences, Nagasaki, Japan) [17,18]. The KDP136 strain was cultured in a BHI medium with 25 μg/mL chloramphenicol, 10 μg/mL erythromycin, and 1 μg/mL tetracycline. *Fusobacterium nucleatum* JCM8532 (type strain) was cultured in a BHI medium in 15 mL centrifuge tubes for 2 days under anaerobic conditions, followed by growing in 50 mL centrifuge tubes. 

### 2.2. OMV Isolation

*Pg*-OMVs, *Pg*KDP-OMVs (OMVs from the KDP136 strain), and *Fn*-OMVs were isolated following an established protocol [8,10]. To evaluate the role of the heat-inactivated gingipains, *Pg*-OMVs were placed in a 70 °C heating block for 1 h (HT*Pg*-OMVs).

### 2.3. Endotoxin Activity Assay

The endotoxin activity level of OMV was determined via using the colorimetric LAL assay (Limulus Color KY Test; Wako Pure Chemical Industries, Osaka, Japan), according to the manufacturer’s instructions. As a positive control, LPS from *Pg* (*Pg*-LPS, InvivoGen, San Diego, CA, USA) was incubated under the same conditions. In the present study, *Pg*-OMVs containing the same level of endotoxin activity as *Pg*-LPS were added to cells to compare the induction of cytokines, using *Pg*-LPS as a positive control. *Fn*-OMVs was matched to the same level of endotoxin activity as *Pg*-LPS for comparison.

### 2.4. DNA Quantification

DNA concentration in purified *Pg*-OMVs were measured using a Qubit 4 Fluorometer (Thermo Fisher Scientific, Waltham, MA, USA) according to the manufacturer’s instructions. 

### 2.5. Cell Culture

The Simian virus-40 antigen-immortalized human gingival epithelial cell line OBA-9 was kindly provided by Prof. Shinya Murakami (Osaka University Graduate School of Dentistry, Osaka, Japan). OBA-9 cells were cultured as described previously [19]. Before the addition of OMVs or *Pg*-LPS, OBA-9 cells were incubated in a medium without bovine pituitary extract and epidermal growth factor for 2 h. 

### 2.6. Quantitative Real-Time Polymerase Chain Reaction (qRT-PCR)

To investigate the effects of OMVs on IL-6 and IL-8 expression, OBA-9 cells were incubated with OMVs or *Pg*-LPS (10–1000 ng/mL) for 3 h. Total RNA was isolated from OBA-9 cells using NucleoSpin RNA (TaKaRa Bio, Otsu, Japan) and cDNA was synthesized from 500 ng of total RNA using PrimeScript RT Master Mix (Perfect Real Time, TaKaRa Bio). qRT-PCR was performed using a CFX96TM Real-Time PCR Detection System (Bio-Rad, Hercules, CA, USA). Template cDNA was mixed with SYBR Green Supermix (Bio-Rad), distilled water, and primers. The reaction was performed at 95 °C for 30 s, followed by 40 cycles at 95 °C for 10 s, and at 60 °C for 30 s. The following primer sets were used: IL-6F, 5′-GCCAGAGCTGTGCAGATGAG-3′; IL-6R, 5′-TCAGCAGGCTGGCATTTG-3′; IL-8F, 5′-ATGACTTCCAAGCTGGCCGTGGCT-3′; IL-8R, 5′-TCTCAGCCCTCTTCAAAAACTTCTC-3′; STINGF, 5′-CCCAGAACATAGACACGCTGGA-3′; STINGR, 5′-ATCAGCTGGGCACCTAGGACA-3′; glyceraldehyde-3-phosphate dehydrogenase (GAPDH)F, 5′-GAGTCAACGGATTTGGTCGT-3′; and GAPDHR, 5′-GACAAGCTTCCCGTTCTCAG-3′. The relative mRNA levels of the target genes were normalized to those of GAPDH as an internal control.

### 2.7. Enzyme-Linked Immunosorbent Assay (ELISA)

Supernatants from OBA-9 cells were collected after 6 h of stimulation with OMVs or Pg-LPS (100 ng/mL). IL-6 and IL-8 protein levels were measured using a sandwich ELISA kit from R&D Systems (Minneapolis, MN, USA), according to the manufacturer’s instructions.

### 2.8. Western Blotting

Cell lysates were subjected to Western blotting, as described previously [19]. The following antibodies were used: rabbit monoclonal anti-phospho-JNK (Thr183/Tyr185, 1:1000 *v/v*); anti-phospho-p38 MAPK (Thr180/Tyr182, 1:1000 *v/v*); anti-phospho-NF-κB p65 (Ser536, 1:1000 *v/v*); anti-NF-κB p65 (1:1000 *v/v*); rabbit polyclonal anti-phospho-Erk1/2 MAPK (Thr202/Tyr204, 1:1000 *v/v*); anti-Erk1/2 (1:1000 *v/v*); anti-JNK (1:1000 *v/v*); and anti-p38 MAPK (1:1000 *v/v*) (all from Cell Signaling Technology, Danvers, MA, USA). Further, Anti-β-actin (1:1000 *v/v*, rabbit monoclonal, Cell Signaling Technology) was used as a protein loading control. 

### 2.9. Inhibition of MAPK and NF-κB Signaling Pathways

To investigate potential pathways contributing to OMV-induced IL-6 and IL-8 expression, OBA-9 cells were either untreated or pre-treated for 1 h with the following inhibitors used at the given concentrations: U0126 (mitogen-activated protein kinase [MEK] inhibitor, 10 µM); SB203580 (p38 MAPK inhibitor, 10 µM); SP600125 (JNK inhibitor, 20 µM); and BAY11-7082 (NF-κB inhibitor, 10 µM) (all from Wako). After inhibitor pre-treatment, OBA-9 cells were cultured with OMVs or *Pg*-LPS (100 ng/mL) for a further 6 h. 

### 2.10. Knockdown of STING mRNA in OBA-9 Cells Using Small Interfering RNA (siRNA)

SiRNA targeting STING, as well as a negative control siRNA (no significant homology to any known mouse, rat, or human gene sequences) were both purchased from Sigma-Aldrich (St. Louis, MO, USA). OBA-9 cells were cultured as described previously [19]. Cells were transfected with siRNA complexes using Lipofectamine^®^ RNAiMAX Reagent (Thermo Fisher Scientific), according to the manufacturer’s instructions. The final concentration of the siRNA transfected into cells was 50 nM. After 24 h, the total RNA and protein were extracted in order to confirm STING knockdown. OBA-9 cells were cultured with *Pg*-OMVs for 3 and 6 h to examine the expression of IL-6 and IL-8 mRNA levels and to measure the levels of these proteins by an ELISA program, respectively. Furthermore, OBA-9 cells were cultured with *Pg*-OMVs or *Fn*-OMVs for 30 min to analyze via Western blotting for the phosphorylation of NF-κB p65. Data were analyzed using the negative control siRNA as a baseline for mRNA knockdown efficiency.

### 2.11. Statistical Analysis

All experiments were performed at least three times to confirm the reproducibility of the results. The results are presented as the mean ± standard deviation (SD) of three independent experiments. The mean values of the groups were compared using a one-way analysis of variance in conjunction with Holm–Sidak’s multiple comparisons test. Statistical significance was set at *p* < 0.05. Further, GraphPad Prism 6.00 software (GraphPad Software, Inc., La Jolla, CA, USA) was used for data analysis. 

## 3. Results

### 3.1. Effect of Pg-OMVs on IL-6 and IL-8 mRNA Levels in OBA-9 Cells

The effect of *Pg*-OMVs, or *Pg*-LPS, on the expression of IL-6 and IL-8 mRNA levels was compared in OBA-9 cells using qRT-PCR. The amount of *Pg*-OMVs added to cells was determined by using the same level of endotoxin activity at each concentration (10, 100, or 1000 ng/mL) of *Pg*-LPS. As shown in Figure 1A, *Pg*-OMVs significantly induced IL-6 and IL-8 mRNA expression at the same levels as endotoxin activity at 10 and 100 ng/mL *Pg*-LPS, whereas *Pg*-LPS significantly promoted the induction of IL-6 and IL-8 mRNA expression in a dose-dependent manner compared to that of the unstimulated controls. *Pg*-OMVs at the same level of endotoxin activity as 100 ng/mL *Pg*-LPS had significantly increased IL-6 and IL-8 mRNA levels compared to that of the unstimulated control (by 6.7-fold and 12.6-fold, respectively) and when also compared to that of 100 ng/mL *Pg*-LPS. *Pg*-OMVs at the same level of endotoxin activity as 1000 ng/mL *Pg*-LPS did not change the expression of IL-6 and IL-8 mRNA when compared to that of the unstimulated control. In fact, many floating dead cells were observed in cells that were cultured with high amounts of *Pg*-OMVs, but this was not observed in cells cultured with *Pg*KDP-OMVs (data not shown). Thus, OBA-9 cells were treated with *Pg*-OMVs at the same level of endotoxin activity as 100 ng/mL *Pg*-LPS in the subsequent experiments. As shown in Figure 1B, *Pg*-OMVs significantly increased IL-6 and IL-8 production in OBA-9 cells. The IL-6 concentration in the supernatant of cells treated with *Pg*-OMVs was significantly higher than that of *Pg*-LPS, whereas the IL-8 concentration levels were similar between *Pg*-OMVs and *Pg*-LPS. 

### 3.2. Pg-OMVs Increased IL-6 and IL-8 mRNA Expression and Protein Production More Than HTPg-OMVs or PgKDP-OMV

To investigate the effect of gingipains in *Pg*-OMVs on the mRNA expression and protein production of IL-6 and IL-8 in OBA-9 cells, qRT-PCR and ELISA were performed. HT*Pg*-OMVs or *Pg*KDP-OMVs significantly increased IL-6 and IL-8 mRNA expression and protein production. However, *Pg*-OMVs had a stronger effect on the induction of IL-6 and IL-8 than those of HT*Pg*-OMVs or *Pg*KDP-OMVs (Figure 2). These results suggested that other than heat-inactivated gingipains in *Pg*-OMVs have the cytokine-inducing activity.

### 3.3. Roles of MAP Kinases in IL-6 and IL-8 Induction by Pg-OMVs

The phosphorylation levels of MAPK proteins in OBA-9 cells treated with *Pg*-OMVs, *Pg*KDP-OMVs, or *Pg*-LPS were analyzed using Western blotting. An increase in Erk1/2, JNK, and p38 phosphorylation was observed within 5 min of treatment with *Pg*-OMVs, *Pg*KDP-OMVs, or *Pg*-LPS, compared to that of the unstimulated control (Figure 3A). To investigate the possible signaling pathways mediating IL-6 and IL-8 induction, OBA-9 cells were pretreated with MAPK inhibitors for 1 h before the addition of *Pg*-OMVs, *Pg*KDP-OMVs, or *Pg*-LPS. U0126 (MEK inhibitor), SP600125 (JNK inhibitor), and SB203580 (p38 inhibitor) significantly downregulated *Pg*-OMV-, *Pg*KDP-OMV-, and *Pg*-LPS-induced IL-6 and IL-8 production (Figure 3B). 

### 3.4. Involvement of NF-κB p65 in IL-6 and IL-8 Induction by Pg-OMVs

To investigate the involvement of NF-κB in OBA-9 cells treated with *Pg*-OMVs, *Pg*KDP-OMVs, or *Pg*-LPS, the phosphorylation level of NF-κB was determined using Western blotting. An increase in NF-κB phosphorylation was observed within 5 min after the stimulation with *Pg*-OMVs or *Pg*KDP-OMVs compared to that of the unstimulated control; however, *Pg*-LPS slightly increased the phosphorylation level in OBA-9 cells at 5 min after the stimulation (Figure 4A). Moreover, the NF-κB inhibitor BAY11-7082 strongly attenuated *Pg*-OMV- or *Pg*KDP-OMV-induced IL-6 and IL-8 production, and *Pg*-LPS-induced IL-6 production (Figure 4B). These results suggested that *Pg*-OMVs activated the MAPK and NF-κB pathways to produce IL-6 and IL-8 by other factors, in addition to gingipains. 

### 3.5. Effect of STING Knockdown on IL-6 and IL-8 Expression 

Next, we examined whether the nucleic acids in *Pg*-OMVs can induce the expression of IL-6 and IL-8 via the STING pathway. In this study, DNA concentrations contained in *Pg*-OMVs ranged from approximately 300–1500 μg/mL. Transfection with STING-specific siRNA significantly inhibited STING mRNA expression (Figure 5A, left panel) and protein levels (Figure 5A, right panel), by more than 65%. STING downregulation via STING-specific siRNA partially suppressed IL-6 and IL-8 mRNA expression as well as protein expression induced by *Pg*-OMVs (Figure 5B,C). These results suggest that the nucleic acid-activating STING pathway induced by *Pg*-OMVs is important for the induction of pro-inflammatory cytokines from human gingival epithelial cells.

### 3.6. Comparison of Pg-OMVs and Fn-OMVs in IL-6 and IL-8 Production by STING Knockdown

Finally, to determine whether OMVs produced by other periodontopathogenic bacteria also activate IL-6 and IL-8 production via the STING-mediated pathway, we compared *Pg*-OMVs and *Fn*-OMVs in OBA-9 cells. The DNA concentration in *Fn*-OMVs was similar to that in *Pg*-OMVs. The amount of DNA in the OMVs was roughly one-tenth of the total protein content. *Fn*-OMVs induced IL-6 and IL-8 production in OBA-9 cells significantly more strongly than *Pg*-OMVs (Figure 6A). Downregulation of STING by STING-specific siRNA significantly suppressed IL-6 production induced by *Pg*-OMVs by 54% and *Fn*-OMVs by 48%; further, IL-8 production was induced by *Pg*-OMVs by 58% and by *Fn*-OMVs by 64% (Figure 6B). These results suggest that the STING pathways activated by nucleic acids is not only specific to *Pg*-OMVs, but also activated by *Fn*-OMVs. Moreover, transfection of OBA-9 cells with STING-specific siRNA resulted in a slight decrease in phosphorylation of NF-κB after stimulation with *Pg*-OMVs or *Fn*-OMVs compared to siRNA controls (Figure 6C). These results suggest that IL-6 and IL-8 production induced by *Pg*-OMVs or *Fn*-OMVs might be partially dependent on the STING pathway by vesicle-associated DNA, thereby resulting in the activation of NF-κB signaling.

## 4. Discussion

*Pg*, which is the most important pathogenic bacterium for periodontitis, produces OMVs [12]. *Pg* sustains a chronic inflammatory state in the host by circulating *Pg*-OMVs into the blood and releasing virulence factors into deep tissues [9]. Moreover, OMVs are stable because they are not affected by host-derived proteases [12]. Therefore, *Pg*-OMVs are thought to affect the pathogenesis and progression of periodontitis and to play roles in systemic diseases related to *Pg* infection [12,20]. *Pg*-OMVs contain several virulence factors, including FimA, LPS, and gingipains, which are key factors associated with chronic periodontitis [7]. However, little is known about the molecular mechanisms underlying the relationship between *Pg*-OMVs and host epithelial cells. 

IL-6 and IL-8 are essential pro-inflammatory cytokines involved in the pathogenesis of periodontitis. IL-6 and IL-8 levels in gingival crevicular fluid are significantly higher in subjects diagnosed with chronic periodontitis than in healthy controls [21]. Therefore, in this study, we investigated the effect of *Pg*-OMVs on the production of IL-6 and IL-8 in gingival epithelial cells and compared this with *Pg*-LPS, which is a key factor in periodontitis development [4]. In our previous study, we showed that the levels of IL-6 and IL-8 expression in OBA-9 cells was increased by *Pg*-LPS [19]. Here, *Pg*-OMVs increased IL-6 and IL-8 mRNA expression and also protein production to be greater than those induced by *Pg*-LPS in OBA-9 cells (Figure 1). These results indicated that *Pg*-OMVs contained both *Pg*-LPS and other virulence factors. Gingipains, a group of proteinases produced by *Pg*, are a significant cause of periodontal tissue destruction [22]. Previous studies revealed a three- to five-fold enrichment of gingipains in OMVs compared to those found within their parent bacteria [23]. Here, *Pg*-OMVs at the same levels as endotoxin activity of 1000 ng/mL Pg-LPS causes cell detachment and changes in the morphology of OBA-9 cells due to the presence of gingipains in *Pg*-OMVs, which is consistent with the results of a previous study [6]. However, changes in the morphology of OBA-9 cells were not observed in cells cultured with HT*Pg*-OMVs or *Pg*KDP-OMVs at the same levels as endotoxin activity of 1000 ng/mL *Pg*-LPS. Therefore, we examined whether HT*Pg*-OMVs or *Pg*KDP-OMVs affected the induction of IL-6 and IL-8 mRNA expression and protein levels in OBA-9 cells. *Pg*-OMVs had a stronger effect on the induction of IL-6 and IL-8 than HT*Pg*-OMVs or *Pg*KDP-OMVs (Figure 2). These results indicated that *Pg*-OMV-associated gingipains were one of the virulence factors that induce IL-6 and IL-8 expression in OBA-9 cells, as well as other remaining factors.

Little is known about the intracellular fate of pathogen OMVs in host epithelial cells. Although previous studies have demonstrated that *Pg*-OMVs penetrate into human oral epithelial cells [11,24], the signaling mechanisms of gingival epithelial cells stimulated with *Pg*-OMVs are not fully understood. OMVs cause the rapid phosphorylation of kinase signaling proteins, NF-κB transactivation, and induction of cytokines [5]. To clarify the possible mechanisms underlying the induction of IL-6 and IL-8 by *Pg*-OMVs in OBA-9 cells, intracellular signaling pathways were explored. Increased phosphorylation of all three major MAPK signaling molecules (Erk1/2, JNK, and p38) in OBA-9 cells was observed within 30 min after either *Pg*-OMV or *Pg*KDP-OMV stimulation. Moreover, pre-treatment with the MEK, JNK, and p38 MAPK inhibitors strongly reduced the production of IL-6 and IL-8 by *Pg*-OMVs or *Pg*KDP-OMVs. These results suggested that Erk1/2, JNK, and p38 participated in the induction of IL-6 and IL-8 expression in OBA-9 cells. In addition, while the NF-κB pathway is a key signaling pathway for the expression of many cytokines and chemokines, NF-κB is also activated by *Pg*-OMVs [25]. In agreement with previous studies, stimulation of *Pg*-OMVs or *Pg*KDP-OMVs increased NF-κB phosphorylation within 15 min after treatment (Figure 4A). The NF-κB inhibitor significantly attenuated the increased IL-6 and IL-8 production by *Pg*-OMVs or *Pg*KDP-OMVs in OBA-9 cells (Figure 4B). These results indicated that *Pg*-OMVs activated Erk1/2, JNK, p38 MAPK, and NF-κB signaling pathways, resulting in increased activation of IL-6 and IL-8 in human gingival epithelial cells.

Inflammatory responses in host cells require the contribution of diverse signaling pathways by various microbial products. To further determine other possible mechanisms underlying the induction of IL-6 and IL-8 by *Pg*-OMVs, STING—which functions as an essential innate immune signaling adaptor molecule that recognizes cytoplasmic cellular or microbial DNA [22]—was examined; this was conducted as *Pg*-OMVs contain DNA. Inflammatory cytokines that induce the cGAS–cGAMP–STING signaling pathway, and are triggered by pathogen-derived DNA, are important in order to protect the host from infection. However, these cytokines can cause tissue damage and immune pathology when produced in excess and sustained. In fact, activation of cGAS–STING has been implicated in the promotion of inflammatory diseases, such as acute pancreatitis [26], acute kidney injury [27], and lung injury [28]. However, activation of the STING pathway in periodontal tissues has not yet been investigated. In this study, STING-specific siRNA suppressed *Pg*-OMV-induced IL-6 and IL-8 mRNA expression (Figure 5B) and markedly reduced concentrations of IL-6 and IL-8 in the cell culture supernatants (Figure 5C). However, *Pg*-OMVs had no effects on STING gene and protein expression in OBA-9 cells (data not shown). Moreover, we found that induction of IL-6 and IL-8 production via the STING pathway was not *Pg*-OMV-specific but also occurred in *Fn*-OMVs (Figure 6B). A previous study has demonstrated that activated STING induces the NF-κB signaling pathway, leading to the induction of pro-inflammatory cytokines [29]. In this study, NF-κB phosphorylation was slightly reduced when stimulated with *Pg*-OMV or *Fn*-OMV in OBA-9 cells with knockdown of STING expression, compared to siRNA controls (Figure 6C). These results suggest that NF-κB-dependent IL-6 and IL-8 expression could be a part of the STING signaling pathway in OBA-9 cells. The cGAS–STING signaling pathway mediates interferon regulatory factor 3 (IRF3)-dependent transcription of type I IFNs and NF-κB-dependent transcription of pro-inflammatory cytokines. It has been suggested that whether STING activates IRF3 or NF-κB differs between species, such as humans and mice, and even between cell types. Furthermore, how STING affects the balance and relative levels of type I IFN expression and NF-κB-dependent inflammatory cytokine production remains to be elucidated [30]. Moreover, whether OMVs predominantly activate the MAPK or STING signaling pathway also remains to be examined. The present study is significant in demonstrating the involvement of the STING pathway in pro-inflammatory cytokine production in periodontal tissues.

## 5. Conclusions

In conclusion, this study demonstrated that *Pg*-OMVs contain a large number of virulence factors and activate complex signaling pathways, including Erk1/2, p38, JNK, STING, and NF-κB, which therefore result in the induction of pro-inflammatory cytokine expression in human gingival epithelial cells (Figure 7). These results suggest that *Pg*-OMVs may play important roles in the pathogenesis and progression of periodontitis by stimulating various pathways. 

## Figures and Tables

**Figure 1 biomedicines-10-02643-f001:**
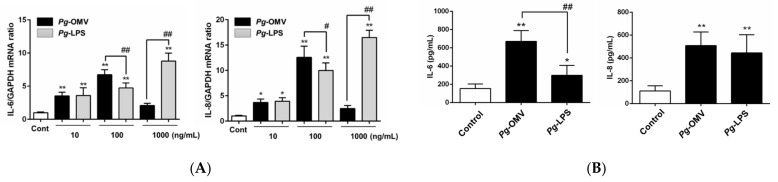
Effect of *Pg*-OMVs or *Pg*-LPS on IL-6 and IL-8 mRNA; additionally, protein expression in OBA-9 cells. (**A**) OBA-9 cells were cultured with *Pg*-OMVs or *Pg*-LPS (10, 100, or 1000 ng/mL) for 3 h. The amount of *Pg*-OMVs added to cells was determined by using the same level of endotoxin activity as each concentration of *Pg*-LPS. IL-6 and IL-8 mRNAs expression levels were analyzed via qRT-PCR. (**B**) OBA-9 cells were cultured with *Pg*-OMVs or 100 ng/mL of *Pg*-LPS for 6 h. IL-6 and IL-8 concentrations in the cell culture supernatants were determined using an ELISA program. Data are the mean ± SD of three independent experiments. * *p* < 0.05 and ** *p* < 0.01, compared to the control. ^#^
*p* < 0.05 and ^##^
*p* < 0.01, compared to *Pg*-OMVs.

**Figure 2 biomedicines-10-02643-f002:**
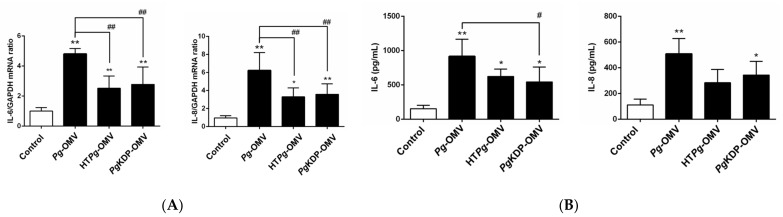
Effect of *Pg*-OMVs, heat-treated (HT)*Pg*-OMVs, or *Pg*KDP-OMVs on IL-6 and IL-8 mRNA and protein expression in OBA-9 cells. (**A**) OBA-9 cells were cultured with *Pg*-OMVs, HT*Pg-*OMVs, or *Pg*KDP-OMVs for 3 h. The amount of OMVs added to cells was determined by using the same level of endotoxin activity as 100 ng/mL of *Pg*-LPS. IL-6 and IL-8 mRNA expression levels were analyzed using qRT-PCR. (**B**) OBA-9 cells were cultured with *Pg*-OMVs, HT*Pg-*OMVs, or *Pg*KDP-OMVs for 6 h. IL-6 and IL-8 concentrations in the cell culture supernatants were determined using an ELISA program. Data are the mean ± SD of three independent experiments. * *p* < 0.05 and ** *p* < 0.01, compared to the control. ^#^
*p* < 0.05 and ^##^
*p* < 0.01, compared to *Pg*-OMVs.

**Figure 3 biomedicines-10-02643-f003:**
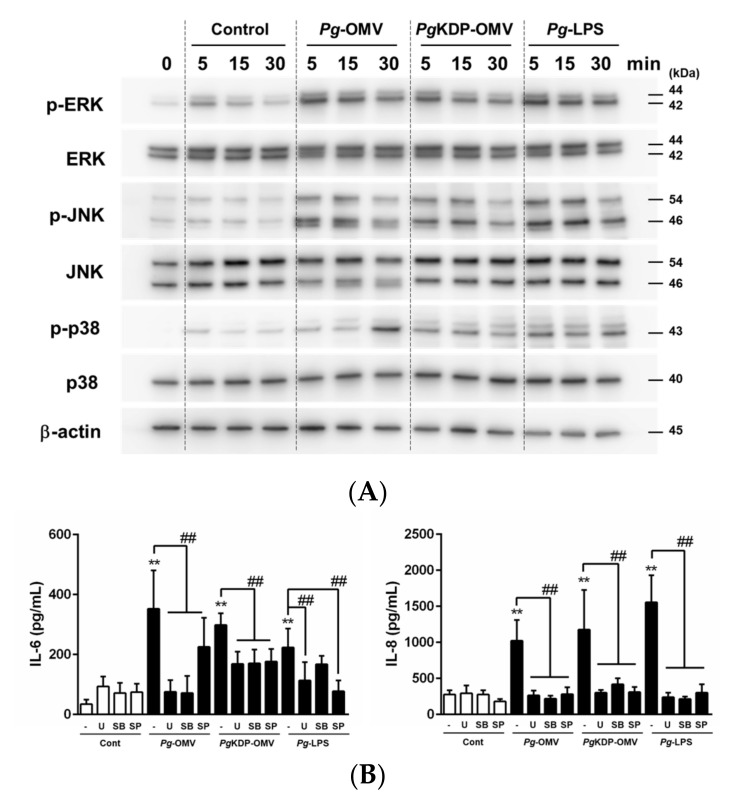
Pathways contributing to the induction of IL-6 and IL-8 production by *Pg*-OMVs, *Pg*KDP-OMVs, or *Pg*-LPS in OBA-9 cells. (**A**) OBA-9 cells were cultured with *Pg*-OMVs, *Pg*KDP-OMVs, or 100 ng/mL of *Pg*-LPS for 5, 15, and 30 min each8. The amount of OMVs added to cells was determined by using the same level of endotoxin activity as 100 ng/mL of *Pg*-LPS. Phosphorylated mitogen-activated protein kinases (MAPKs) (Erk1/2, JNK, and p38) were assessed by Western blotting. Representative blots are shown. (**B**) OBA-9 cells were pre-treated with U0126 (10 μM), SP600125 (20 μM), or SB203580 (10 μM) for 1 h before the addition of *Pg*-OMVs, *Pg*KDP-OMVs, or 100 ng/mL of *Pg*-LPS, for 6 h. IL-6 and IL-8 concentrations in the cell culture supernatants were determined using an ELISA program. Data are the mean ± SD of three independent experiments. ** *p* < 0.01 compared to the unstimulated control. ^##^
*p* < 0.01 compared to the control without inhibitor.

**Figure 4 biomedicines-10-02643-f004:**
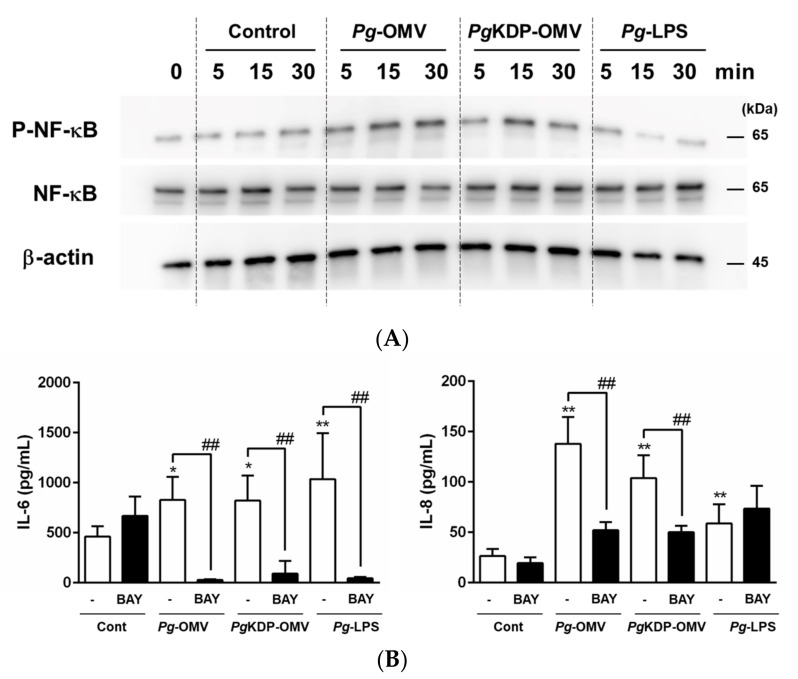
Involvement of NF-κB in the induction of IL-6 and IL-8 production by *Pg*-OMVs, *Pg*KDP-OMVs, or *Pg*-LPS in OBA-9 cells. (**A**) OBA-9 cells were cultured with *Pg*-OMVs, *Pg*KDP-OMVs, or 100 ng/mL of *Pg*-LPS for 5, 15, and 30 min each. The amount of OMVs added to cells was determined by using the same level of endotoxin activity as 100 ng/mL of *Pg*-LPS. Phosphorylated NF-κB was assessed by Western blotting; representative blots are shown. (**B**) OBA-9 cells were pre-treated with BAY11-7082 (10 μM) for 1 h before the addition of *Pg*-OMVs, *Pg*KDP-OMVs, or 100 ng/mL of *Pg*-LPS, for 6 h. IL-6 and IL-8 concentrations in the cell culture supernatants were determined using an ELISA program. Data are the mean ± SD of three independent experiments. * *p* < 0.05 and ** *p* < 0.01, compared to the unstimulated control. ^##^
*p* < 0.01 compared to the control without inhibitor.

**Figure 5 biomedicines-10-02643-f005:**
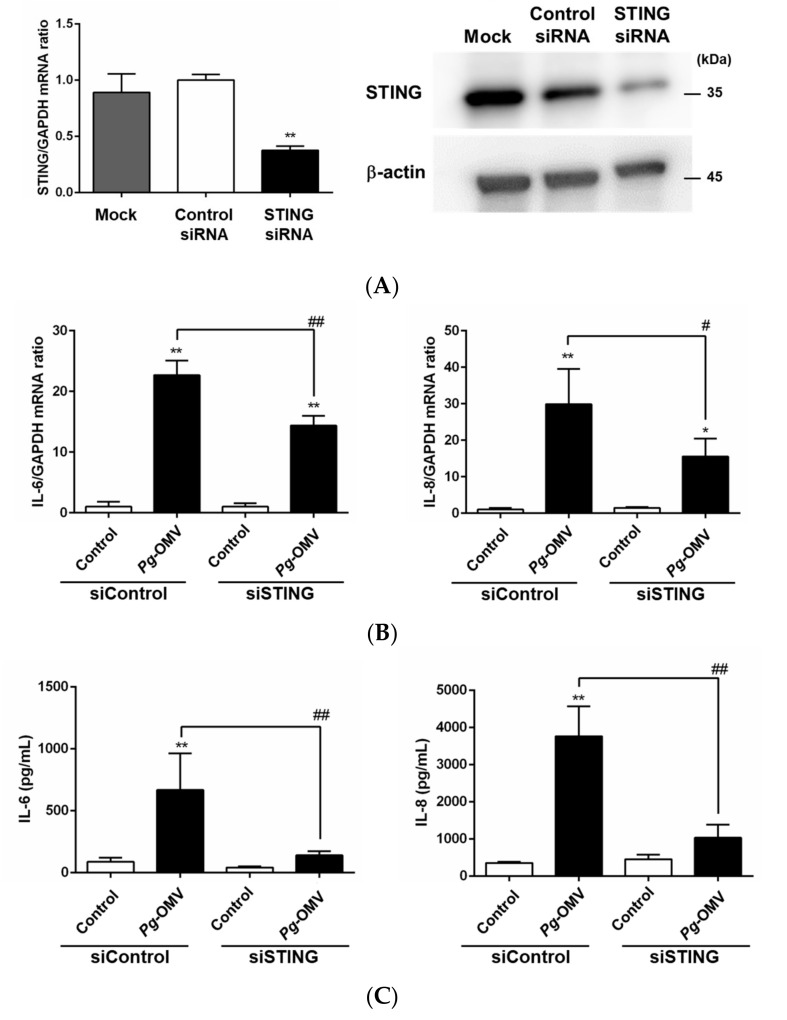
The effect of STING knockdown on the mRNA expression and protein levels of IL-6 and IL-8 in OBA-9 cells. OBA-9 cells were transfected with STING siRNA for 24 h (**A**); and cultured with *Pg*-OMVs for 3 h (**B**); and then 6 h (**C**). The amount of *Pg*-OMVs added to cells was determined by using the same level of endotoxin activity as 100 ng/mL of *Pg*-LPS. (**A**, **left**) STING mRNA was analyzed using qRT-PCR. (**A**, **right**) STING protein levels were determined by Western blotting; representative blots are shown. (**B**) IL-6 and IL-8 mRNA expression levels were analyzed using qRT-PCR. (**C**) IL-6 and IL-8 concentrations in the cell culture supernatants were determined using an ELISA program. Data are the mean ± SD of three independent experiments. * *p* < 0.05 and ** *p* < 0.01, compared to the unstimulated control. ^#^
*p* < 0.05 and ^##^
*p* < 0.01, compared to the negative control siRNA.

**Figure 6 biomedicines-10-02643-f006:**
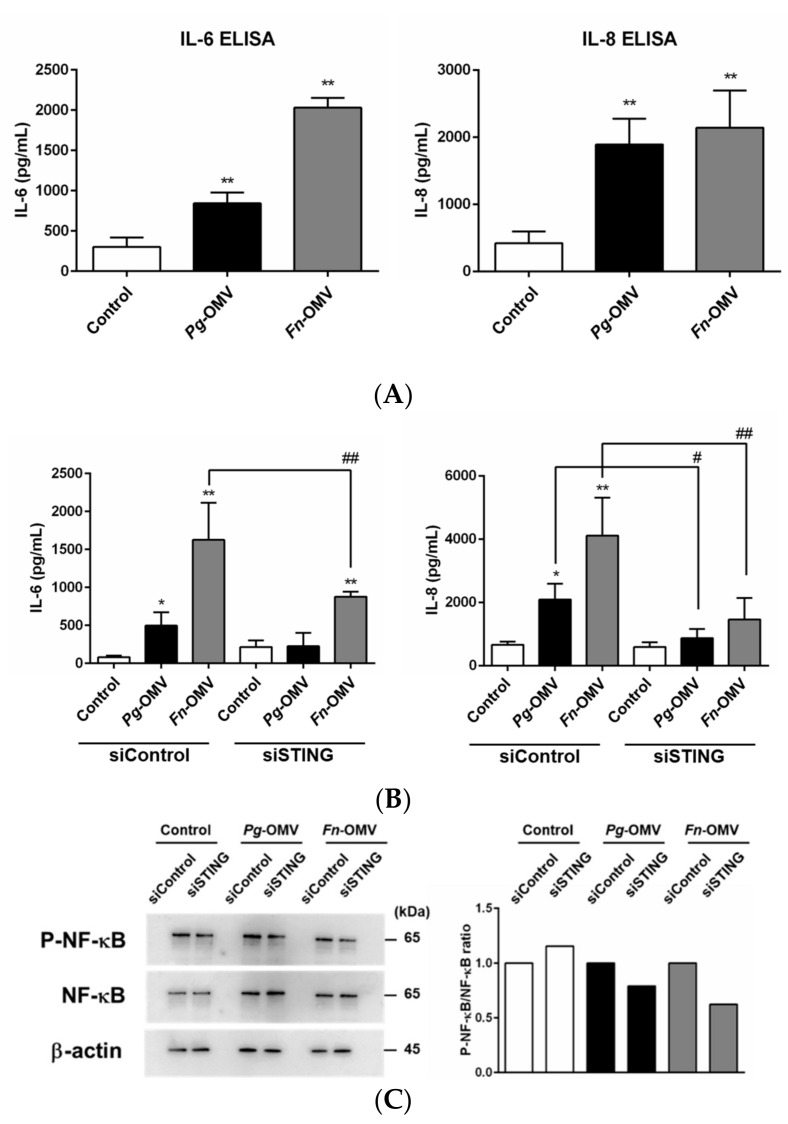
Comparison of *Pg*-OMVs and *Fn*-OMVs in IL-6 and IL-8 production by STING knockdown. (**A**) OBA9-cells were cultured with *Pg*-OMVs or *Fn*-OMVs for 6 h. The amount of OMVs added to cells was determined by using the same level of endotoxin activity as 100 ng/mL of *Pg*-LPS. IL-6 and IL-8 concentrations in the cell culture supernatants were determined using an ELISA program. (**B**,**C**) OBA-9 cells were transfected with STING siRNA for 24 h and cultured *Pg*-OMVs or *Fn*-OMVs for 6 h (**B**) and 30 min (**C**). (**B**) IL-6 and IL-8 concentrations in the cell supernatants were determined using an ELISA program. Data are the mean ± SD of three independent experiments. * *p* < 0.05 and ** *p* < 0.01, compared to the unstimulated control. ^#^
*p* < 0.05 and ^##^
*p* < 0.01, compared to the negative control siRNA. (**C**) NF-κB p65 protein levels were determined by Western blotting; representative blots are shown. The ratio of p-NF-κB and NF-κB is shown after correction with β-actin by densitometric analysis. That same ratio for Si control was set to 1.0.

**Figure 7 biomedicines-10-02643-f007:**
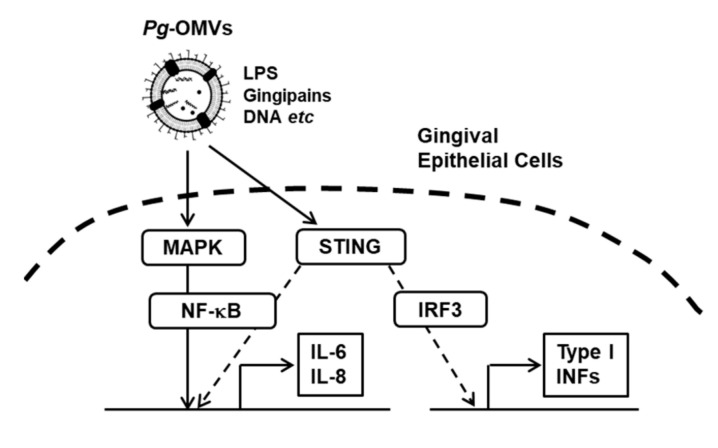
A schematic diagram showing the proposed pathway in *Pg*-OMV-induced IL-6 and IL-8 expression in human gingival epithelial cells. *Pg*-OMVs contain various virulence factors, such as LPS and gingipains. *Pg*-OMVs activate Erk1/2, JNK, p38 MAPK, STING, and NF-κB signaling pathways, leading to increased IL-6 and IL-8 expression in human gingival epithelial cells.

## Data Availability

Data is contained within the article.

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
