# Peer review of "Porphyromonas gingivalis Outer Membrane Vesicles Stimulate Gingival Epithelial Cells to Induce Pro-Inflammatory Cytokines via the MAPK and STING Pathways"

_biomedicines, 2022, doi:10.3390/biomedicines10102643_

Round 1
Reviewer 1 Report
It requires major modifications.
Author Response
Reply to the comment of Reviewer 1:
We appreciate you reviewing our manuscript. According to the other reviewer’s suggestions, we have carefully revised the manuscript.
Reviewer 2 Report
The paper Biomedicines-1897289 is a Research Article aimed to give information on the molecular mechanisms involved in Pg-OMVs-induced inflammatory response of gingival epithelial cells. The authors observed that Pg-OMVs were able to induce IL-6 and IL-8 production in OBA-9 cells by activating Erk1/2, JNK, p38 MAPK, STING, and NF-κB signaling pathways.
The results are interpreted appropriately and the conclusions sufficiently supported by the results.
However, minor revisions are required to make the paper suitable for publication in the Special Issue of Biomedicines "Immune Response to Viruses and Bacteria".
-Figure 3B does not report the results described in the text (IL-6 and IL-8 production in the presence of MAPK inhibitors), and it is identical to Figure 4B. The authors must substitute Fig. 3B with a new one showing the results obtained with cells pretreated with the inhibitors.
-Page 8, line 280 (legend of Fig. 5): please, delete “mRNAs and protein levels”
Author Response
Reply to the comments of Reviewer 2:
We appreciate you reviewing our manuscript. We have carefully read your comments and revised the manuscript as follows.
- We appreciate your pointing this out. As reviewer pointed out, Figure 3B was incorrect and did not reflect the results. We have changed Figure 3B with a correct one showing the results obtained with cells pretreated with MAPK inhibitors.
- According to the reviewer’s suggestion, we have deleted the words “mRNAs and protein levels” in legend of Figure 5 (Page 8, line 281).
Reviewer 3 Report
This manuscript described a novel role of Porphyromonas gingivalis outer membrane vesicles in stimulating the gingival epithelial cells to induce pro-inflammatory cytokines via the MAPK and STING pathways. The authors used Pg-OMVs, PgKDP-OMVs and Fn-OMVs to compare the expression of IL-6 and IL-8 in human gingival epithelial cells. They show that Pg-OMVs induced IL-6 and IL-8 production via the extracellular signal-regulated kinase (Erk) 1/2, c-Jun N-terminal kinase (JNK), p38 MAPK, and NF-κB signaling pathways in OBA-9 cells. Some suggestions are listed below.
1. What is the rational to compare the effect of Pg-OMVs on IL-6 and IL-8 levels in OBA-9 cells ? Any clinical data or reports to support the hypothesis ? Do the authors had a chance to compare other pro-inflammatory cytokines like IL-1beta, TNF-alpha and IL-18 or others?
2.For results Figure 6, author mentioned STING-specific siRNA resulted in a slight decrease in phosphorylation of NF-κB after stimulation with Pg-OMVs or Fn-OMVs compared to siRNA controls. And in Figure 7, a schematic diagram showing the STING is the upstream of NF-κB. And NF- κB p65 protein levels were determined by western blotting in Figure 6. But the Figure 6c data did not show any significant decreased of NF-κB in STING-specific siRNA groups to my understand.
Author Response
Reply to the comments of Reviewer 3:
We appreciate you reviewing our manuscript. We have carefully read your comments and revised the manuscript as follows.
- IL-6 and IL-8 have been well characterized as major players in chronic periodontitis and shown to be related to the risk and pathogenesis of periodontitis. In our previous study, we showed that the levels of IL-6 and IL-8 expression in OBA-9 cells was increased by Pg-LPS stimulation (Hiroshima Y et al., J Cell Biochem. 2018, 119, 1591–1603, Supporting Information). In clinical study, IL-6 and IL-8 levels in gingival crevicular fluid are significantly higher in subjects diagnosed with chronic periodontitis than in healthy controls (Stadler AF et al., J Clin Periodontol. 2016, 43(9):727–45). Therefore, the present study focused on the levels of IL-6 and IL-8 induced by the stimulation with Pg-OMVs compared to Pg-LPS.
We also investigated the effect of Pg-OMVs on the levels of IL-1b in OBA-9 cells by real-time PCR and ELISA. However, no significant increase in IL-1b production was observed upon Pg-OMVs stimulation. We have described these points and included a new reference in the revised manuscript (Page 10, lines 327-333).
(New Reference)
- Stadler, A.F.; Angst, P.D.; Arce, R.M.; Gomes, S.C.; Oppermann, R.V.; Susin, C. Gingival crevicular fluid levels of cyto-kines/chemokines in chronic periodontitis: a meta-analysis. J Clin Periodontol. 2016, 43, 727-45. DOI:10.1111/jcpe.12557.
- As the reviewer pointed out, Figure 6C shows that the phosphorylation of NF-kB stimulated by Pg-OMVs or Fn-OMVs seemed to be decreased, but may not be significantly different, in OBA-9 cells with knockdown of STING expression, compared to siRNA controls. NF-kB-dependent IL-6 and IL-8 expression could be part of the STING signaling pathway in OBA-9 cells. It has been reported that the possibility that the cGAS-STING signaling pathway mediates interferon regulatory factor 3 (IRF3)-dependent transcription of type I interferon (IFN)s and NF-kB-dependent transcription of pro-inflammatory cytokines (Chin E.N. et al., Trends Cell Biol. 2022). It has been suggested that whether STING activates IRF3 or NF-kB differs between species, such as humans and mice, and even between cell types. Furthermore, how STING affects the balance and relative levels of type I IFN expression and NF-kB-dependent inflammatory cytokine production remains to be elucidated. The present study is significant in demonstrating the involvement of the STING pathway in pro-inflammatory cytokine production in periodontal tissues. We have described these points and included a new reference in the revised manuscript (Page 11, lines 390-401). We have also changed the schematic diagram in Figure 7.
(New Reference)
- Chin, E.N.; Sulpizio, A.; Lairson, L.L. Targeting STING to promote antitumor immunity. Trends Cell Biol. 2022 Aug 2:S0962-8924(22)00149-0. DOI:10.1016/j.tcb.2022.06.010.
Round 2
Reviewer 3 Report
I do not have any further concern about this revised version.